# Chinese Character Component Deformation Based on AHP

**Tian Chen [1,2], Fang Yang [1,2,*] and Xiang Gao [1,2]**

[1] School of Cyber Security and Computer, Hebei University, Baoding 071002, China
[2] Hebei Machine Vision Engineering Research Center, Hebei University, Baoding 071002, China
* Correspondence: yangfang@hbu.edu.cn

**Abstract:** Since Chinese characters are composed of components, deforming the components in a small number of existing calligraphy characters to generate new characters is an effective method to produce a Chinese character library in the same style. Usually, the component deformation is achieved by affine transformation. However, when calculating the parameters in affine transformation, existing methods usually have the problems of a large amount of manual participation or complicated calculation. In this paper, we proposed an Analytic Hierarchy Process (AHP)-based Chinese character component deformation method, which is simple in calculation and can effectively realize the deformation of Chinese character components on the basis of reducing manual intervention. We first determined the factors that affect the selection of control points in affine transformation, then used AHP to calculate the weights of feature points and select the control points according to the weights. As a prerequisite for affine transformation, a matching method of Chinese character feature points based on the Chinese character skeleton map and neighborhood information is also proposed, which helps to achieve more efficient deformation. Experimental results on different fonts demonstrate the effectiveness and generality of our method.

**Keywords:** Chinese character component deformation; affine transformation; AHP; feature matching

## 1. Introduction

Currently, with the widespread application of computer fonts, there are many types of online Chinese character libraries. More and more people prefer to communicate with others in their preferred writing style rather than a uniform printing style, such as some personal handwriting fonts, traditional calligraphy fonts, etc. However, there are 3500 commonly used characters, and the official standard GB2312 for the Chinese character set has a total of 6763 Chinese characters. It is almost impossible for a calligrapher to write so many Chinese characters in the same style correctly. Therefore, how to automatically generate more characters from a small number of existing Chinese characters on the basis of retaining the original writing style has attracted the interest of many researchers.

One possible way is the utilization of deep learning techniques. In recent years, deep learning has performed well in style transfer and image generation [1,2]. Many people use it to solve Chinese character problems, usually by learning the writing style of the target font to convert the print font into the target font [3,4]. However, the characters obtained by such methods often differ from the real ones in structure and style. That is mainly because deep learning-based approaches are good at the transfer of image texture and color, but Chinese characters have complex structures. Even small changes in the location and geometry of their elements (e.g., components, and strokes) can dramatically change their meanings or styles.

A valuable characteristic of Chinese characters is that they are highly hierarchical and a lot of them are composed of basic components [5,6]. It is worth noting that the number of Chinese character components is far less than the number of Chinese characters. According to [7], there are only 514 components segmented from 3500 commonly used characters. As shown in Figure 1, the same component can form different Chinese characters. Another

efficient way to solve the problem is to deform the existing components to generate new components that can be used to compose a new character. It can also be seen from Figure 1 that the shape of the same component varies with its position in Chinese characters, which is usually realized by an affine transformation. This is mainly because in the deformation of Chinese character components, it is necessary to ensure the fluency of strokes and the invariance of structure, and affine transformation has the property of ensuring the straightness and parallelism of images. However, a common difficulty we often encounter while manipulating affine transformations is how to calculate the parameters. Existing methods usually have a large amount of manual participation or complicated calculations. For example, Feng et al. [8] proposed a component deformation method based on affine transformation, but the control points need to be selected manually, and the amount of manual participation is large. Zu et al. [9] calculated affine transformation parameters by constructing nonlinear equations, but the result may have multiple solutions, which need to be verified one by one.

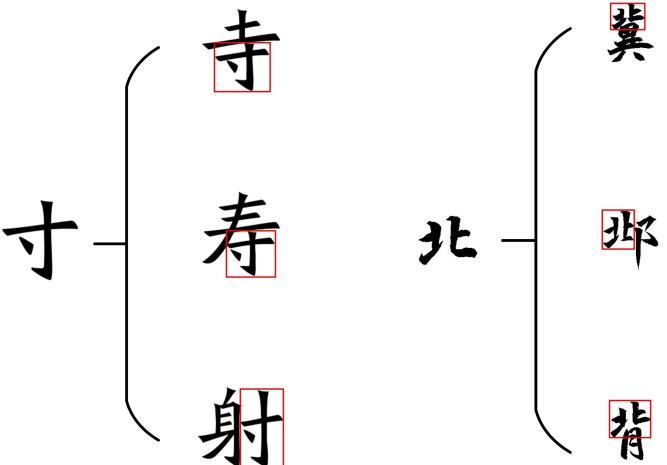

**Figure 1.** One component can be deformed to form different Chinese characters.

In order to achieve a more efficient and high-quality deformation of Chinese character components, we proposed an Analytic Hierarchy Process (AHP)-based affine transformation parameter calculation method, which can effectively realize the deformation of Chinese character components on the basis of reducing manual intervention and simple calculation. Both qualitative and quantitative results in different fonts prove that the method we proposed can achieve a better effect. The overall procedure is shown in Figure 2.

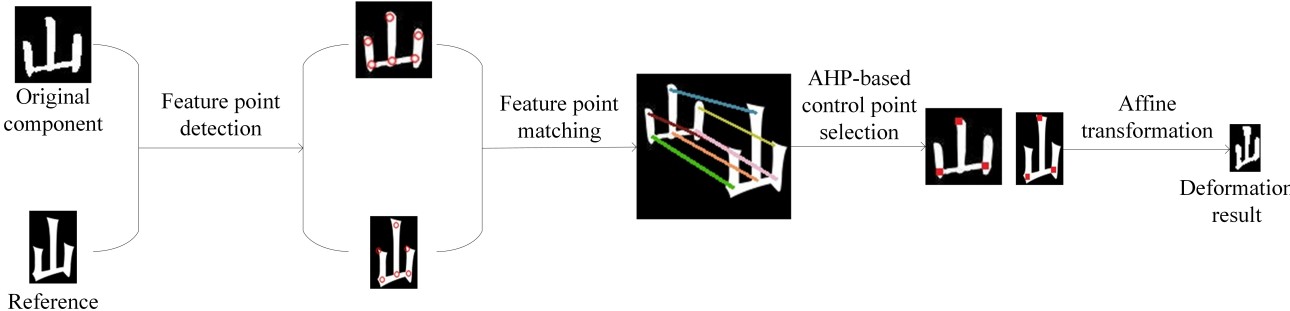

**Figure 2.** The overall procedure of our experiment. The input consists of an original component image and a reference component image. The original component is a handwritten font, and the reference component is a regular print font that provides a reference for the deformation of the original component. The final output is a new component image that retains the original writing style.

Major contributions of this study are twofold:

(1) A feature point matching method is proposed. As a prerequisite for affine transformation, we designed an effective Chinese character feature point matching method, which extracts feature points from the Chinese character skeleton map and matches them according to their neighborhood information.

(2) An automatic control point selection technology is proposed. First, we determined the factors that affect the selection of control points, then grouped the feature points, and used AHP to calculate the weight of each group of feature points. The group with the highest weight is used as the optimal control points for affine transformation parameter calculation.

## 2. Related Works

### 2.1. Feature Point Detection and Matching

Currently, there are many highly respected feature matching methods in the field of image processing, such as Harris corners, Shi-Tomas corners, SIFT, SURF, ORB, and GMS.

Harris [10] determines the feature points according to the change in gray value in the sliding window and uses the idea of the autocorrelation function. Shi-Tomas [11] is an improvement of Harris. It differs in the selection of the final discriminant and enhances the stability of corner detection. The methods in [12,13] have strong stability and are not disturbed by image rotation and scale scaling, but the calculation process is complex and the running time is long. ORB [14] improves the calculation speed, but there are often many matching errors in practical applications. GMS [15] obtains high-quality matches by eliminating false matching pairs. However, these methods are usually suitable for image with obvious color texture, and the effect is not ideal in Chinese character feature point matching.

### 2.2. Chinese Character Component Deformation

Existing methods for the deformation of Chinese character components can be divided into three categories. One is to start from the strokes and formulate transformation rules. This method usually makes mistakes in the structure of Chinese characters and has a small scope of application. For example, Liu et al. [16] proposed a deformation technique based on the skeleton diagram of characters, which established a graphical model for strokes and generated deformation by isomorphic triangles and computational interpolation. Sun et al. [17] designed a transformation sequence for each stroke and realized the distortion-free scaling transformation of Chinese characters, but only for the Song type.

Another one is to convert other font styles into the font we need. For example, Wu et al. [18] proposed a method to change Chinese characters from "Regular Script" to "Semi-Cursive Script" by using the Trail Point Set description. Lian et al. [19] generated shape templates for characters, used the Coherent Pointment Drift (CPD) algorithm to implement point set registration, and finally achieved the deformation of Chinese characters from SimHei to KaiTi with shape interpolation. However, the Chinese characters obtained by this method usually have blurred and deformed outlines.

Another deformation method is to deform the component as a whole by affine transformation. For example, Liu et al. [20] achieved the deformation of the Chinese character component by establishing a control relationship between the skeleton point and the contour point. However, some manual intervention is still required in the process. The method in [21,22] transformed the image function to establish a linear equation system about the parameters of the affine transformation, but it is powerless for binary images. Yun et al. [23] selected three groups of generalized centroid points to calculate the affine transformation parameters, but the three points may be on the same straight line, resulting in no solution. There are also some algorithms based on local optimization, such as Moving Least Squares [24], but when used in Chinese characters, the phenomenon of stroke distortion often occurs.

Based on the above problems, we proposed an AHP-based affine transformation control point selection method. This paper is organized as follows: In Section 3, we detail the feature point matching algorithm. In Section 4, we introduce the detailed application of AHP. In Section 5, we present the results of our method. In Section 6, we discuss our proposal and future work directions.

### 3. Feature Point Matching

#### 3.1. Feature Point Detection

The foundational part of our feature matching work sought to find the representative feature points. As a special kind of figure, the characteristic points of Chinese characters include endpoints, inflection points, and intersection points. As shown in Figure 3, different feature points have different neighborhood information on the skeleton map. There are only two foreground pixels in the eight-neighborhood of a common point. There is one foreground pixel and three foreground pixels in the eight-neighborhood of the endpoint and the intersection point, respectively. For the inflection point, we usually take the neighborhood of $\left( \frac{w}{10} \times \frac{h}{10} \right)$, and there is a certain angle inside. w and h represent the width and height of the character, respectively.

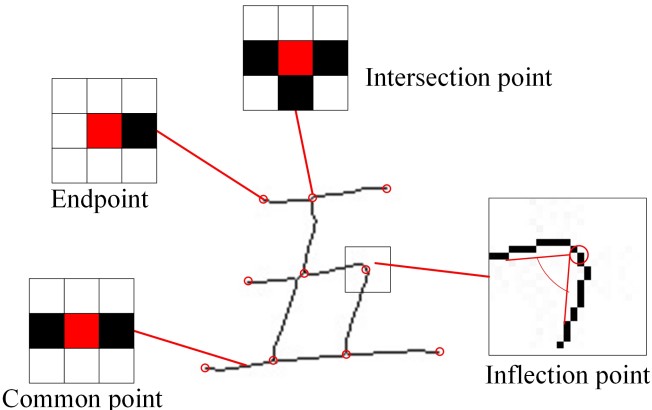

**Figure 3.** Different neighborhood features for different feature points.

The feature points found using this method often appear redundant (Figure 4) and need to be filtered by nonmaximal suppression. In the same condition, we show our method can identify the feature points more accurately in Figure 5.

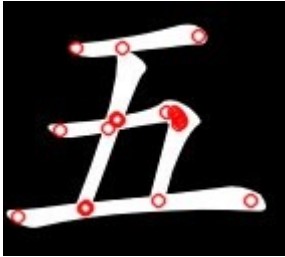

**Figure 4.** Feature points found by neighborhood information.

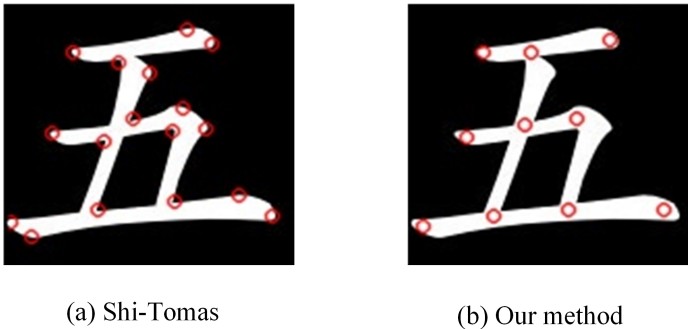

(a) Shi-Tomas          (b) Our method

**Figure 5.** With the same nonmaximum suppression, (**a**) is the feature point detected result using the Shi-Tomas Corners, and (**b**) is the result using our method. This proves that our method can perform feature point detection accurately.

*3.2. Feature Point Matching*

As shown in Figure 6, when a component is used to form different characters, the relative position of the feature points does not change. Thus, we perform an initial match by calculating the angle between each feature point and the component centroid. The result is shown in Figure 7.

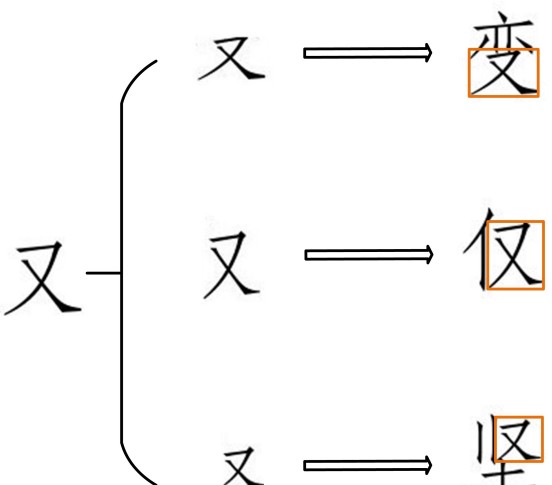

**Figure 6.** When the components are used to form different characters, the relative position of the feature points does not change.

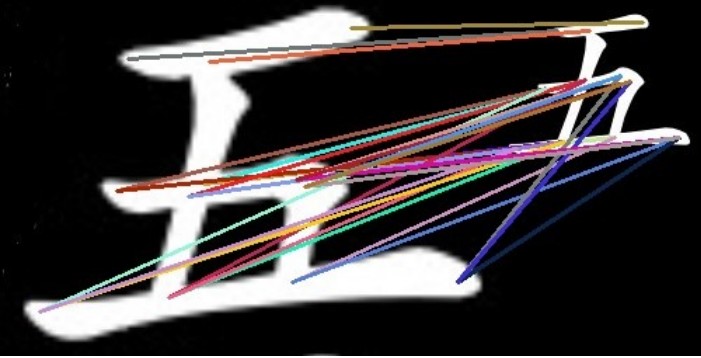

**Figure 7.** Feature point matching by the angle between each feature point and the character centroid.

As we can see, there are many false matches because the angle of the correct matching point will also have slight differences, and we set an error tolerance value. Therefore, we borrowed the idea of GMS and used neighborhood features to filter the false matches.

Given a pair of images taken from different views, if the neighboring pixels move together, GMS implies that a true match shares many similar features in the same region across both images. In contrast, false matches view different regions and have far fewer similar features. This idea is also applicable to Chinese character feature matching.

We first extract the feature point neighborhood based on the aspect ratio of the character, i.e., the aspect ratio of the extracted region is the same as that of the Chinese character. In this study, we extract regions by one-third of the width of Chinese characters. Then, as shown in Figure 8, the extracted area is divided into 16 grids. By calculating the ratio of the number of foreground pixels to the total number of pixels in each grid, we can obtain a 16-dimensional feature vector. We use the Euclidean distance to calculate the similarity between the two vectors and filter the matching pairs. In other words, for a feature point with multiple matching points, the matching point with the highest degree of similarity is reserved. Our final matching result is shown in Figure 9.

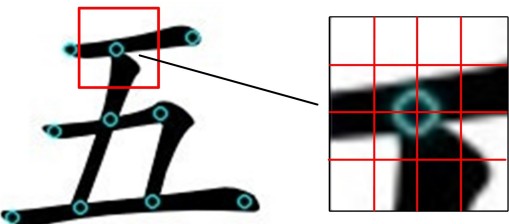

**Figure 8.** Information about the Feature Point neighborhood.

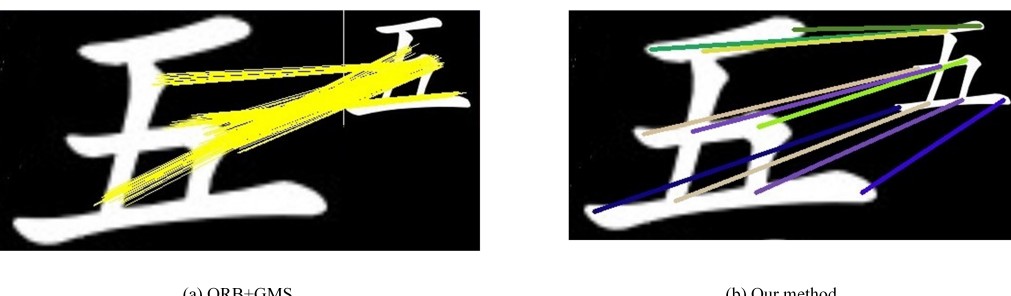

(a) ORB+GMS          (b) Our method

**Figure 9.** ORB and GMS have difficulty in this case because the color and texture features of Chinese characters are not obvious. Our method can perform accurate feature point detection and eliminate false matches with neighborhood information.

## 4. Control Point Selection and Component Deformation

The general model for affine transformations is :

$$\begin{cases} X = a_1 x + b_1 y + c_1 \\ Y = a_2 x + b_2 y + c_2 \end{cases} \tag{1}$$

Some rules for determining control points in the transformation of Chinese character components are proposed in [8,17]. We summarized them and consider the process of control point selection as a multicriteria decision-analysis problem [25]. An effective way to solve this type of problem is the Analytic Hierarchy Process (AHP) [26]. This is a method to model and quantify the human decision-making process. The profound theoretical foundation and simple expression make it favored by many researchers [27]. It usually consists of three steps:

(1)     Establishing the hierarchical structure model;
(2)     Hierarchical single ranking and consistency test;
(3)     Hierarchical total ranking and consistency test.

### 4.1. Hierarchical Structure Model

For the deformation of the Chinese character component, the width and height often change the most before and after the transformation, so they should be considered as point selection criteria. If the control points are selected too close together, this will usually result in a skewed and distorted image after the transformation, which should also be taken into account when selecting the points.

The final hierarchy structure is shown in Figure 10. H and W represent the width and height properties, respectively. When we divide an image into a 3×3 grid, N is the number of grids covered by triangles formed by the points in the point set. For statistical purposes, we specify that a grid that is covered by more than one-third of the area is marked as covered.

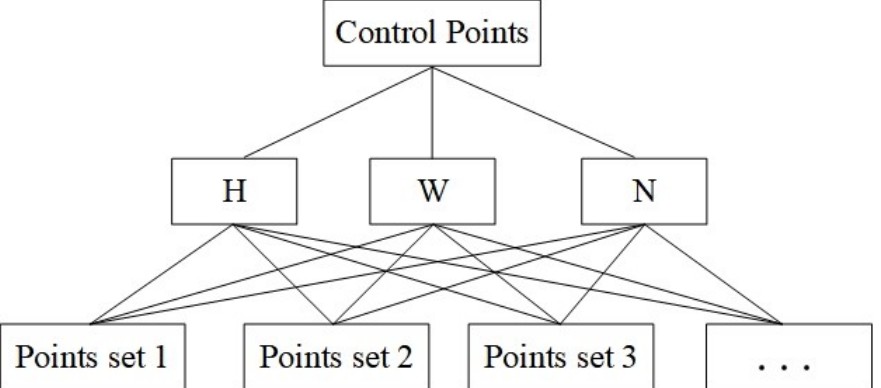

**Figure 10.** Hierarchical structure. The top level is the target level, the middle level is the criterion level, and the bottom level is the scheme level.

### 4.2. Hierarchical Single Ranking

#### 4.2.1. Criterion Layer Weights

First, we need to construct the judgment matrices of H, W, and N and then determine the weight of each. In general, the judgment matrix needs to be constructed by expert scoring, which is usually highly subjective. It may fail when performing the consistency tests mentioned later. Thus, the importance of the factors in criterion level is obtained by comparing the difference in width and height between the original component and the reference component in this study.

Set $\Delta w$ and $\Delta h$ to represent the difference in width and height, respectively. We use r to denote the ratio of the larger to the smaller of $\Delta w$ and $\Delta h$, rounded up. We stipulate that N has the same importance as the larger one between W and H.

Suppose the importance of W relative to H is $\rho$, and the importance of N relative to W is $\varphi$. Then, the value of $\rho$ and $\varphi$ need to satisfy Equation (2).

$$\begin{cases} \rho = r, \varphi = 1 (\Delta w > \Delta h) \\ \rho = \frac{1}{r}, \varphi = r(\Delta w < \Delta h) \end{cases} \tag{2}$$

Thus, the judgment matrix of influencing factors is shown in Table 1.

**Table 1.** Importance matrix of criterion level.

|   | **W** | **H** | **N** |
|---|---|---|---|
| W | 1 | $\rho$ | $\frac{1}{\varphi}$ |
| H | $\frac{1}{\rho}$ | 1 | $\frac{1}{\rho\varphi}$ |
| N | $\varphi$ | $\rho\varphi$ | 1 |

The distribution of weights for W, H, and N by column normalization using the arithmetic mean method is shown in Equation (3). $n$ represents the order of the matrix, and $\omega_i$ represents the weight value of the $i$th element.

$$\omega_i = \frac{1}{n} \sum_{j=1}^{n} \left( \frac{a_{ji}}{\sum_{k=1}^{n} a_{kj}} \right) \tag{3}$$

4.2.2. Scheme Level Weights

Define $S = \{p_1, p_2, \ldots p_m\}$ as the set of feature points of the component. Every three points in S are grouped, for a total of $C_m^3$ groups. The number of point sets obtained in this way may be large, so we need to filter them. The rules are as follows:

If any two points $p_i, p_j (i = A, B, C; j = A, B, C; i \neq j)$ in the points set $P = (p_A, p_B, p_C)$ satisfy Equation (4) or Equation (5), or the number of grids covered by a triangle formed by the three points exceeds 4, it is kept; otherwise, it is deleted. $\sigma$ and $\varepsilon$ are the acceptable tolerance ranges.

$$\left| p_i.x - p_j.x \right| = w \pm \sigma \tag{4}$$

$$\left| p_i.y - p_j.y \right| = h \pm \varepsilon \tag{5}$$

After that, the judgment matrix of the feature point sets under each criterion factor needs to be constructed by their importance. The degree of importance is quantified by the relative scale of the pairwise comparison of the two elements. Table 2 is the importance scale proposed by [26].

**Table 2.** Importance scale.

| Degree of Importance | Value |
|---|---|
| Equally Important | 1 |
| Slightly Important | 3 |
| Strongly Important | 5 |
| Particular Important | 7 |
| Extremely Important | 9 |
| Median value of two adjacent judgments | 2 4 6 8 |

The values shown in Table 2 are usually set manually, with strong subjectivity, and there are usually problems that cannot pass the consistency check mentioned later. We have made the rules for this comparison. Taking W as an example, first, we calculate the error from the component width for each point set according to Equation (6) and denote it by $\alpha$. $p_x, p_y (x = a, b, c; y = a, b, c; x \neq y)$ are any two points in the point set $X = (p_a, p_b, p_c)$, w is the width of the component.

$$\alpha = w - max \left( \left| p_x.x - p_y.x \right| \right) \tag{6}$$

It can be seen that the smaller the value of $\alpha$, the higher the importance of the corresponding point set. Therefore, we compare the importance of the two point sets by $\alpha$. For any two point sets, T and Q, we use I to denote the importance of T relative to Q. The value of I is shown in Equation (7).

$$\begin{cases} I = 1 \ (-t \leq \alpha_T - \alpha_Q < t) \\ I = \frac{1}{2} \left( t \leq \alpha_T - \alpha_Q < 2t \right) \\ I = 2 \left( -2t \leq \alpha_T - \alpha_Q < -t \right) \\ \ldots \end{cases} \tag{7}$$

Through experiments on four different Chinese character datasets (Italic, Fangsong, Lishu, and SimHei) with image sizes of $141 \times 155$, we conclude that the value of t is preferably between 1–5; otherwise, it may lead to an inaccurate selection of control points.

The construction method of the judgment matrix under the H attribute is the same as that of the W. The judgment matrix under the N attribute can be directly obtained by calculating the ratio of the number of covered grids. After the judgment matrix of the points set is obtained, the weights are calculated by Equation (3).

### 4.2.3. Consistency Test

Since the judgment matrices we constructed are all positive reciprocal matrices, and their maximum eigenvalues are $\lambda \geq n$, (where n is the order of the matrix), this may lead to unreasonable matrices. For example, in Table 3, $a_{12} = 3$ and $a_{23} = \frac{1}{2}$, which leads to $a_{13} = \frac{3}{2}$, while the result according to the importance comparison rule is $a_{13} = 2$. Therefore, we need to judge whether this irrationality is acceptable or not through a consistency test.

**Table 3.** Importance scale.

|       | Set A | Set B | Set C | Set D | Set E |
|-------|-------|-------|-------|-------|-------|
| Set A | 1     | 3     | 2     | 1     | 2     |
| Set B | 1/3   | 1     | 1/2   | 1/3   | 1/2   |
| Set C | 1/2   | 2     | 1     | 1/2   | 1     |
| Set D | 1     | 3     | 2     | 1     | 2     |
| Set E | 1/2   | 2     | 1     | 1/2   | 1     |

The calculation of the consistency index is shown in Equation (8). $\lambda$ is the maximum characteristic value of the importance matrix. According to the definition of the maximum characteristic root of the matrix: $A\omega = \lambda\omega$, $\lambda$ can be calculated from Equation(9).

$$CI = \frac{\lambda - n}{n - 1} \tag{8}$$

$$\lambda = \sum_{i=1}^{n} \frac{[A\omega]_i}{n\omega_i} \tag{9}$$

When $CI = 0$, the matrix has perfect consistency, and the larger the $CI$, the higher the inconsistency. To measure the value of $CI$, the random consistency index $RI$ and the consistency ratio $CR$ are introduced. The value of $RI$ can be calculated by the definition of the average random consistency index [28], and the values of $RI$ are shown in Table 4 when the order of the judgment matrix is 1–15.

**Table 4.** Stochastic consistency index RI value.

| Matrix Order | 1 | 2 | 3 | 4 | 5 | 6 | 7 | 8 | 9 | 10 | 11 | 12 | 13 | 14 | 15 |
|--------------|---|---|------|------|------|------|------|------|------|------|------|------|------|------|------|
| *RI* | 0 | 0 | 0.56 | 0.89 | 1.12 | 1.26 | 1.36 | 1.41 | 1.46 | 1.49 | 1.52 | 1.54 | 1.56 | 1.58 | 1.59 |

The consistency ratio is defined as Equation (10), and when $CR < 0.1$, the inconsistency degree of the judgment matrix is considered to be within the permissible range and passes the consistency test. Conversely, when $CR \geq 0.1$, the inconsistency degree is considered to be outside the permissible range and fails the consistency test. Since our matrix is generated by a fixed rule comparison, the subjectivity is greatly reduced compared with the traditional construction methods. Our method greatly improves the pass rate of the consistency check.

$$CR = \frac{CI}{RI} \tag{10}$$

### 4.3. Hierarchical Total Ranking

After calculating the weights of the point sets relative to each criterion, we need to calculate the total weight of each point set. The process is shown in Equation (11). $\omega_{i\_T}$

represents the final weight of the ith point set, $\omega_{i\_W}$, $\omega_{i\_H}$, and $\omega_{i\_N}$ represent the weights of the ith point set under *W*, *H*, and *N*, relatively.

$$\omega_{i\_T} = \omega_{i\_W} * \omega_W + \omega_{i\_H} * \omega_H + \omega_{i\_N} * \omega_N \tag{11}$$

Similarly, the total hierarchical ranking needs to be tested for consistency, and the process is shown in Equation (12). Where $CI_W$, $CI_H$, and $CI_N$ represent the corresponding calculated *CI* values under the W, H, and N attributes.

$$\begin{cases} CI_T = \omega_W * CI_W + \omega_H * CI_H + \omega_N * CI_N \\ RI_T = \omega_W * RI_W + \omega_H * RI_H + \omega_N * RI_N \\ \qquad CR_T = \frac{CI_T}{RI_T} \end{cases} \tag{12}$$

*4.4. Component Deformation*

After hierarchical total ranking, we can find the point set with the highest weight and then find the corresponding points in the reference component for the points in the highest weight point set by using the feature point matching results mentioned before. These points will be used as control points for the calculation of the parameters in Equation (1). Finally, the new component image can be obtained by an affine transformation of the original component image. The overall process is shown in Algorithm 1.

---

**Algorithm 1** AHP-based Chinese character component deformation.

---

**Input:** Original component image $T_1$, reference component image $T_2$
**Output:** The component image $T_R$ obtained by deforming $T_1$.

1: Rosenfeld $(T_1, T_2)$;//obtain the skeleton map of the input image.
2: MatchPairs<-Feature point detection and matching;
3: A(W,H,N);//Compute the importance matrix A of (W, H, N) according to Equation (2).
4: Weight(W,H,N);//Compute $\omega_W$, $\omega_H$, and $\omega_N$ according to Equation (3).
5: PoSets<-Group(points);//Grouping the original component feature points in groups of three.
6: PoSets<-Filter(PoSets);//Filter the sets of feature points obtained in step 5.
7: **for** C *in* [W,H,N] **do**
8:     B(PoSets);//Compute the importance matrix B of the feature point sets.
9:     **for** pset *in* PoSets **do**
10:         Weight(pset);
11:     **end for**
12:     Consistency test(B);//Consistency test according to Equation (10).
13: **end for**
14: **for** pset *in* PoSets **do**
15:     Total-Weight(pset);//Compute the total weight of each point set according to Equation (11).
16: **end for**
17: CPoints<-Control-Points(PoSets, MatchPairs);//Determine control points.
18: Parameters<-Calculation(CPoints);//Calculating affine transformation parameters.
19: $T_R$<-Affine-transformation($T_1$, Parameters);

---

## 5. Results and Analysis

*5.1. Evaluation Metrics*

SSIM is used to measure the structural similarity between the real image and the generated image. It takes the value of [0,1], and the larger value indicates the higher similarity between the two images. The formula is expressed as:

$$SSIM(R, F) = \frac{(2\mu_R\mu_F + c_1)(2\sigma_R\sigma_F + c_2)}{(\mu_R^2 + \mu_F^2 + c_1)(\sigma_R^2 + \sigma_F^2 + c_2)} \tag{13}$$

where R and F indicate the real image and the generated image, respectively. $\mu_R$ and $\mu_F$ are the pixels mean of R and F, respectively. $\sigma_R^2$ is the pixel variance of x, $\sigma_F^2$ is the pixel variance of y. $c_1$, $c_2$, and $c_3$ are constants to avoid a denominator of 0. In this study, we set $c_1 = (k_1 L)^2$, $c_2 = (k_2 L)^2$, and $c_3 = \frac{c_2}{2}$. Generally, $k_1 = 0.01$, $k_2 = 0.03$.

*5.2. Results*

The main development tool for the experiment is Visual Studio 2017, and the programming language is C++. The system environment is Microsoft Windows 10.

5.2.1. Results of Our Method

According to [7], there are 3500 commonly used characters and 311 commonly used components. Among them, there are 63 components, each of which can constitute 1% or more of the number of commonly used Chinese characters. In this study, we selected 20 components with a large number of characters and a large morphological variation on different characters. The effectiveness of the method in this study is tested in the case of generating components with different forms by affine transformation of the same component several times. The selected components and the deformation times, as well as the corresponding number of SSIM mean values, are shown in Table 5.

**Table 5.** Number of component deformations and the corresponding SSIM mean values.

| Components | Times | SSIM Mean Values |
| :---: | :---: | :---: |
| 日 | 7 | 76.78% |
| 田 | 5 | 74.58% |
| 虫 | 5 | 75.23% |
| 比 | 4 | 67.12% |
| 巾 | 4 | 78.37% |
| 酉 | 3 | 74.01% |
| 力 | 3 | 82.80% |
| 龙 | 2 | 65.98% |
| 口 | 5 | 81.77% |
| 目 | 5 | 77.36% |
| 山 | 3 | 71.80% |
| 土 | 4 | 73.57% |
| 米 | 2 | 76.78% |
| 又 | 4 | 78.49% |
| 寸 | 3 | 69.51% |
| 母 | 3 | 78.32% |
| 瓦 | 3 | 72.39% |
| 石 | 3 | 78.16% |
| 贝 | 3 | 80.28% |
| 白 | 3 | 79.17% |

It can be seen that the SSIM between the real characters and the transformed ones can reach 82.80%, and the average SSIM value is about 75.91%.

In order to verify the effectiveness of using our method in practical applications, we randomly selected 60 Chinese character components from the "Si Ku Quan Shu", a publicly recognized reference document for the study of ancient Chinese characters. When dealing with ancient Chinese characters, there may be cases where the amount of data is not enough to find the required components. Therefore, it is necessary to use similar fonts as reference components to deform the original components with affine transformation. According to the writing characteristics of "Si Ku Quan Shu", we choose "Song" as the reference component. Eventually, the SSIM value of these characters can reach up to 80.07%, and the average SSIM is about 72.40%.

### 5.2.2. Comparison with MLS

MLS [24] is a classical algorithm in the field of image deformation and has great applications in many research areas. The basic idea of MLS is to control deformation through source and target control points. Therefore, in the case of using the same feature point matching algorithm, we compared our method with MLS.

We randomly selected several Chinese characters from four common fonts and split them into components. Due to the simple structure of some components, the shape changes are not obvious when forming characters, and the result is of little significance. We selected some components with a larger morphological change for experiments. The SSIM is calculated to verify the generalizability of the method we proposed.

The SSIM values obtained by our method and the MLS for the deformation of the four fonts are shown in Table 6. As can be seen, the SSIM values obtained by our method on all four fonts are higher than those obtained by the MLS, which further demonstrates the effectiveness of our method.

**Table 6.** Different fonts deformation data.

| Font | Quantity of Components | Average SSIM Value of MLS | Average SSIM Value of Our Method |
|---|---|---|---|
| Kaiti | 70 | 54.18% | 73.07% |
| SimHei | 51 | 60.17% | 78.67% |
| Fangsong | 55 | 58.70% | 74.75% |
| Lishu | 43 | 62.69% | 72.27% |

We also tested on other fonts and the effect of using our method and MLS is shown in Figure 11. It can be seen that the results obtained using the MLS are highly distorted in terms of strokes or structure. Some of them change the structure, such as ▨ . Some have a writing style that differs too much from the original component, such as ⧖. Some have too much local deformation, such as 由, 寸 . The fourth row in Figure 11 shows the components obtained using our method, which, visually speaking, preserves the original writing style and structural features of the original component well. The Chinese character in the last row shows the new characters composed by stitching together the already existing components and the components generated by our method, which have basically the same style and can be a good fit. Therefore, from a subjective point of view, our generated components can be better applied to the Chinese character generation task.

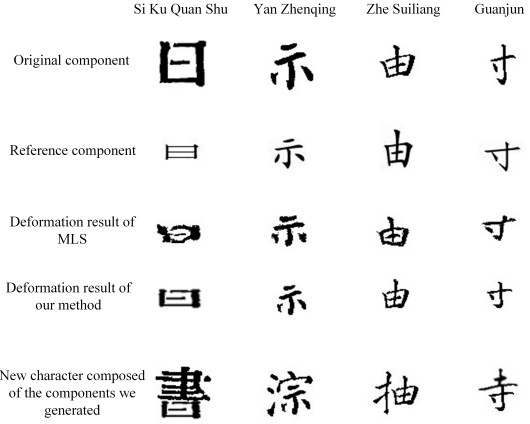

**Figure 11.** From left to right, the components are from Si Ku Quan Shu, Yan Zhenqing, Zhe Suiliang, and Guanjun. From top to bottom are the original components used for deformation, the reference components during deformation, the components generated by MLS, the components generated by our methods, and the characters containing the new components we generated.

## 6. Conclusions

In this study, we proposed an affine transformation control point selection method based on AHP, which aims to realize the deformation of Chinese character components. We first use the neighborhood information to achieve feature point matching. Then, we transform the control point selection problem into a multicriteria decision-analysis problem and solve it with AHP. Finally, the component is deformed using an affine transformation. Both quantitative and qualitative results demonstrate the effectiveness of our method by comparing it with MLS under the same conditions. However, there will still be inconsistencies in the judgment matrix of some complex components. In the future, further research will be conducted on how to efficiently create or adjust the judgment matrix.

**Author Contributions:** All authors contributed to this work. Writing-original draft preparation, T.C.; writing-review and edit, F.Y.; supervision, X.G. All authors have read and agreed to the published version of the manuscript.

**Funding:** This research was funded by Science and Technology Project of Hebei Education Department (ZD2019131) and "one province, one university" fund of Hebei University (No. 521000981155).

**Data Availability Statement:** Not applicable.

**Conflicts of Interest:** The authors declare no conflict of interest.

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
