# Peer review of "Chinese Character Component Deformation Based on AHP"

_applsci, doi:10.3390/app121910059_

Round 1

Reviewer 1 Report

From image processing point of view, the methods are rather standard and intuitive. There are several heuristics without  theoretical analysis, as the authors admit in the text (for instance, on p. 8 they wrote: "Through experiments .... we conclude that the value of t is preferably between 1-5". )

The method was developed particularly for Chinese characters and uses their specific properties ("A valuable characteristic of Chinese characters is that they are highly hierarchical, and a lot of them are composed of basic components.") The method is probably not transferable to any other character set. 

Since I do not speak/read Chinese, it is impossible for me to evaluate the application aspect of the paper.   I do not know whether or not the proposed method is of practical relevance in Chinese character processing. 

Reviewer 2 Report

The paper presents techniques to improve working with Chinese characters' components in different fonts when generating written characters. 

While, in general, the paper presents many details about your workflow, with feature points selection and affine transformation, I felt uneasy understanding exactly the problem you are solving. I understand that you have components from which you generate the character and that you have to transform these components to create new characters. However, it is unclear how to define the "feature points", or if I can call them skeleton points for your shape and then how you use that information for affine transformation. To make the article clearer, I would suggest the following:

1) Outline the method using either a diagram or the algorithm, stating what the input for your system is (is it a character? is it a source component and a target character?), what the output is and what exact steps you do in-between. 

2) Select a case study and present it in a way it shows your method from the input to the output. You show some results in figure 10, but for someone not writing with Chinese characters, these results do not make much sense without context. Are characters you generate meaningful?

3) Consider asking the native speaker to check your text for grammar and the narrative. The text is rather hard to read in some parts, and some words like "matrixes" would clearly benefit from grammar correction. 

Overall, I think the paper is rather niche, and to make it appealing to a wider audience and improve its readability, it would benefit from revision. 

Reviewer 3 Report

Authors have proposed an affine transformation control point selection method 252 based on AHP, which aims to realize the deformation of Chinese character components. 253 We first use the neighborhood information to achieve feature point matching.

1. Authors have selected  20 commonly used components. Need more information about total number of components that are commonly used

2. They have achieved~78% accuracy. Need more explanation why author believes this number is good

3. Need more brief explanation on methodology
